# MULTI-GRAINED ENTITY PROPOSAL NETWORK FOR NAMED ENTITY RECOGNITION

## ABSTRACT

In this paper, we focus on a new Named Entity Recognition (NER) task, i.e., the Multi-grained NER task. This task aims to simultaneously detect both fine-grained and coarse-grained entities in sentences. Correspondingly, we develop a novel Multi-grained Entity Proposal Network (MGEPN). Different from traditional NER models which regard NER as a sequential labeling task, MGEPN provides a new method that proposes entity candidates in the Proposal Network and classifies entities into different categories in the Classification Network. All possible entity candidates including fine-grained ones and coarse-grained ones are proposed in the Proposal Network, which enables the MGEPN model to identify multi-grained entities. In order to better identify named entities and determine their categories, context information is utilized and transferred from the Proposal Network to the Classification Network during the learning process. A novel Entity-Context attention mechanism is also introduced to help the model focus on entity-related context information. Experiments show that our model can obtain state-of-the-art performance on two real-world datasets for both the Multi-grained NER task and the traditional NER task.

## 1 INTRODUCTION

In order to really understand the semantic meanings of natural languages, the first step is to identify meaningful entities from the raw text effectively. Such a process is usually called Named Entity Recognition (NER), which is one of the fundamental tasks in natural language processing (NLP). A typical NER takes an utterance as the input and outputs identified entities, such as person names, locations, and organizations. The extracted named entities can benefit various subsequent NLP tasks, including syntactic parsing (Koo & Collins, 2010), question answering (Krishnamurthy & Mitchell, 2015) and relation extraction (Lao & Cohen, 2010). However, accurately recognizing representative entities remains challenging.

Recent work treats NER as a sequence labeling problem. For example, Peters et al. (2017) achieves the state-of-the-art performance on sequence labeling problem by incorporating deep recurrent neural networks (RNN) (Hochreiter & Schmidhuber, 1997) with conditional random field (CRF) (Lafferty et al., 2001). However, a critical problem that arises by treating NER as a sequence labeling task is that it can only recognize non-overlapping entities. It fails to detect entities when they are over-lapping with each other: e.g. a fine-grained entity and a coarse-grained entity as shown in Figure 1. Traditional NER approaches can only identify either "Belgain Grand Prix" or "Grand Prix", but not both of them unless two NER models with different granularities are trained seperately. Here, we defined the NER task that needs to detect both fine-grained entities and coarse-grained entities as the Multi-grained Named Entity Recognition (MGNER) problem.

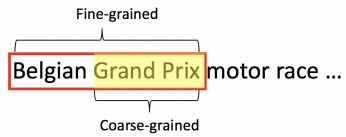

Figure 1: An example from the CoNLL-2003 dataset. In this utterance, the coarse-grained named entity "Grand Prix" overlaps with the fine-grained named entity "Belgian Grand Prix".

To solve the MGNER problem, we propose a novel deep neural network named Multi-grained Entity Proposal Network (MGEPN). Essentially, the idea of MGEPN is to first propose entity candidates and then classify these candidates into different categories. The proposed MGEPN model is composed of two sub-modules: the Proposal Network that proposes entities and the Classification Network that classifies entities. In the Proposal Network, all possible entity candidates including fine-grained ones and coarse-grained ones will be proposed like (Lee et al., 2017b; He et al., 2018). Consequently, our model is able to detect both fine-grained entities and coarse-grained entities without explicitly assuming that entities are non-overlapping or totally nested (Finkel & Manning, 2009). To improve the model performance as well as to speed up the model learning process, context information learned in the Proposal Network is transferred into the Classification Network. In the Classification Network, a novel Entity-Context attention mechanism is proposed to help the model focus on entity-related context for classification.

In summary, the contributions of this work are:

- We study the Multi-grained Named Entity Recognition (MGNER) problem, which aims to detect both coarse-grained and fine-grained name entities in a single system.
- We propose a novel deep neural network, the Multi-grained Entity Proposal Network (MGEPN), which is composed of a Proposal Network and a Classification Network for MGNER.
- MGEPN utilizes context information to detect entities boundaries in the Proposal Network, which is transferred to the Classification Network during the NER process. In the Classification Network, a novel Entity-Context attention mechanism is proposed to help the model concentrate on entity-related context information for entity type classification.

## 2 PROPOSED MODEL

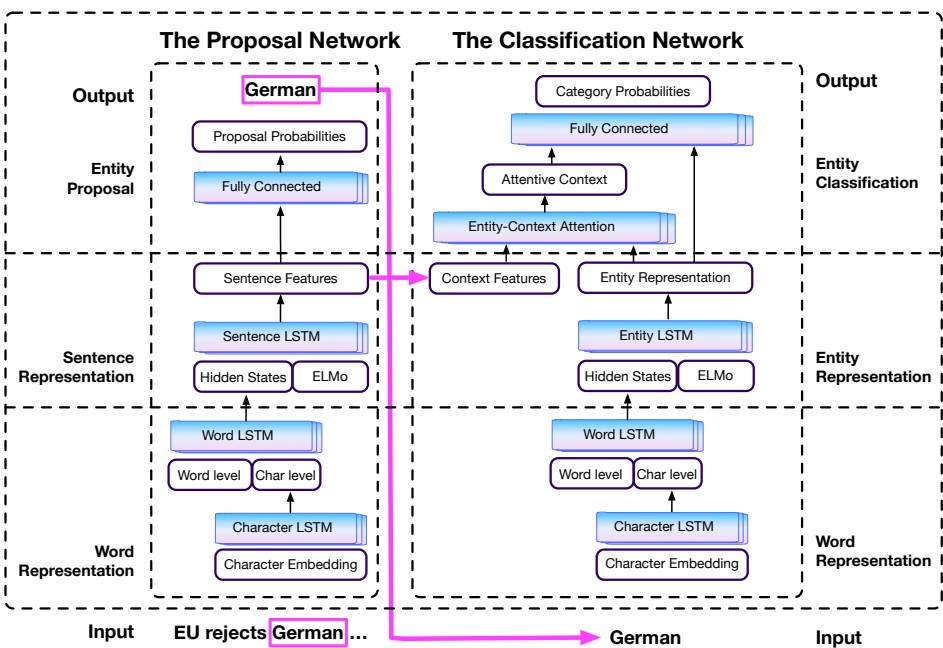

Figure 2: The framework of the Multi-grained Entity Proposal Network (MGEPN). It consists of a Proposal Network and a Classification Network.

To detect both fine-grained entities and coarse-grained entities, we propose a novel NER approach, called Multi-grained Entity Proposal Network (MGEPN). The overall framework of MGEPN is illustrated in Figure 2. Specifically, MGEPN consists of two modules: the Proposal Network and the Classification Network. The Proposal Network is a deep neural network that proposes potential named entities including fine-grained entities and coarse-grained entities based on an input utterance. The second module, Classification Network, aims at classifying entities proposed in the Proposal Network into pre-defined entity categories. Sentence-level semantic features are first trained in the Proposal Network and then transferred into the Classification Network to introduce and utilize the

context information. A novel attention mechanism is also involved in the proposed MGEPN model with a special focus on the Entity-Context relationship. It helps the model capture and utilize entity-related context information. It is worthy to mention that in order to improve the learning speed as well as the model performance of MGEPN, the Proposal Network and the Classification Network are trained with a series of shared input features, including the pre-trained word embeddings and the pre-trained language model features.We provide the key building blocks and the properties of the Proposal Network in Section 3.1 and the Classification Network in Section 3.2, respectively.

## 2.1 THE PROPOSAL NETWORK

The Proposal Network aims at generating possible entity proposals for each utterance. It takes an utterance as input and outputs a set of entity candidates. Essentially, we use a semi-supervised neural network inspired by Peters et al. (2017) to model this process. The architecture of the Proposal Network is illustrated in the left part of Figure 2. Three major components are contained in the Proposal Network: Word Representation, Sentence Representation and Entity Proposal. More specifically, pre-trained word embeddings and character-level word information are used for generating semantic meaningful word representations. Word representations and the language model embeddings, ELMo (Peters et al., 2018), are concatenated together to produce context aware sentence representations. For the entity proposal task, possible named entities are proposed, and sentence representations are fed into a fully connected layer to output the probability of a proposal being a real entity.

### 2.1.1 WORD REPRESENTATION

Given an input utterance with $K$ tokens $(t_1, ..., t_K)$, each token $t_k(1 \leq k \leq K)$ is represented as $\mathbf{x}_k$ using a concatenation of a word-level embedding $\mathbf{w}_k$ and a character-level word information $\mathbf{c}_k$: $\mathbf{x}_k = [\mathbf{w}_k; \mathbf{c}_k]$. The pre-trained word embedding $\mathbf{w}_k$ with dimension $D_w$ is obtained from GloVe (Pennington et al., 2014), and the character-level word information $\mathbf{c}_k$ is obtained with a bidirectional LSTM (Hochreiter & Schmidhuber, 1997) layer to capture the morphological information. The hidden size of this character LSTM is set as $D_{cl}$. As shown in the bottom of Figure 2, character embeddings are fed into the character LSTM. Those character embeddings are randomly initialized and learned within the Proposal Network. The final hidden states from the forward and backward character LSTM are concatenated as the character-level word information $c_k$.

### 2.1.2 SENTENCE REPRESENTATION

To learn the context information from each sentence, another bidirectional LSTM, named word LSTM, is applied to sequentially encode the utterance. For each token, the forward hidden states $\overrightarrow{\mathbf{h}}_k$ and the backward hidden states $\overleftarrow{\mathbf{h}}_k$ are concatenated into the hidden states $\mathbf{h}_k$. The dimension of the hidden states of the word LSTM is set as $D_{wl}$.

$$\overrightarrow{\mathbf{h}}_k = \text{LSTM}_{fw}(\mathbf{x}_k, \overrightarrow{\mathbf{h}}_{k-1}), \ \ \overleftarrow{\mathbf{h}}_k = \text{LSTM}_{bw}(\mathbf{x}_k, \overleftarrow{\mathbf{h}}_{k+1}), \ \ \mathbf{h}_k = [\overrightarrow{\mathbf{h}}_k; \overleftarrow{\mathbf{h}}_k]. \tag{1}$$

Besides, we also utilize the language model embeddings pre-trained in the Elmo (Peters et al., 2018) fashion which is an unsupervised way. The pre-trained Elmo embeddings are concatenated to the hidden states $\mathbf{h}_k$ in the word LSTM. Hence, the concatenated hidden states $\mathbf{h}_k$ for each token can be reformulated as:

$$\mathbf{h}_k = [\overrightarrow{\mathbf{h}}_k; \overleftarrow{\mathbf{h}}_k; \mathbf{ELMo}_k], \tag{2}$$

where $\mathbf{ELMo}_k$ is the ELMo embeddings for token $t_k$. A three-layer hierarchy bi-LSTM neural network is trained as the language model. Since the lower-level LSTM hidden states have the ability to model syntax properties and higher-level LSTM hidden states can capture context information, Elmo computes the language model embeddings as a weighted combination of all the bidirectional LSTM hidden states:

$$\mathbf{ELMo}_k = \gamma \sum_{l=0}^{L} u_j \mathbf{h}_{k,l}^{LM}, \tag{3}$$

where $\gamma$ is a task-specified scale parameter which indicates the importance of the entire ELMo vector to the NER task. $L$ is the number of layers used in the pre-trained language model, the vector $\mathbf{u} = [u_0, \cdots, u_L]$ represents softmax-normalized weights that combines different layers. $\mathbf{h}_{k,l}^{LM}$ are the language model hidden states of layer $l$ at time step $k$.

A sentence bidirectional LSTM layer with hidden dimension $D_{sl}$ is employed on the top of the concatenated hidden states $\mathbf{h}_k$. The forward and backward hidden states in this sentence LSTM are concatenated for each token as the final representation $\mathbf{f}_k \in \mathbb{R}^{2D_{sl}}$.

### 2.1.3 ENTITY PROPOSAL

Using the semantic meaningful features obtained in $\mathbf{f}_k$, we can identify representative entity proposals for each utterance. The strategy of finding entity candidates is to first generate all possible entity proposals and then estimate the probability of being an entity or not for each proposal.

Different types of entity proposals are generated surrounding each token position, and a sliding window is applied among the sequence to output all possible ones. In order to limit the number of generated proposals, we set the maximum length of an entity proposal to $R$. Thus, there will be at most $R$ types of different entity proposals generated at each position. An example of how the proposed method generates different types of entity proposals for the token $t_3$ is illustrated in Figure 3. Specifically, assuming that an input utterance consisting of a sequence of five tokens $(t_1, t_2, t_3, t_4, t_5)$, and the Proposal Network takes a maximum length of proposals $R$ of 5. Here at token $t_3$, we employ the following five proposals: $(t_3), (t_3, t_4), (t_2, t_3, t_4), (t_2, t_3, t_4, t_5), (t_1, t_2, t_3, t_4, t_5)$.

| | | | | | |
|---|---|---|---|---|---|
| **Proposal 1:** | $t_1$ | $t_2$ | $t_3$ | $t_4$ | $t_5$ |
| **Proposal 2:** | $t_1$ | $t_2$ | $t_3$ | $t_4$ | $t_5$ |
| **Proposal 3:** | $t_1$ | $t_2$ | $t_3$ | $t_4$ | $t_5$ |
| **Proposal 4:** | $t_1$ | $t_2$ | $t_3$ | $t_4$ | $t_5$ |
| **Proposal 5:** | $t_1$ | $t_2$ | $t_3$ | $t_4$ | $t_5$ |

Figure 3: All possible entity proposals generated surrounding token $t_3$ when the maximum length of an entity proposal $R$ is set as 5.

For each sliding-window location (token), we simultaneously estimate the probability of a proposal being an entity or not, for all types of entity proposals. A fully connected layer with a two-class softmax function is used to determine the quality of entity proposals:

$$\mathbf{s}_k = \text{softmax}\left(\mathbf{f}_k \mathbf{W}_p + \mathbf{b}_p\right), \tag{4}$$

where $\mathbf{W}_p \in \mathbb{R}^{2D_{sl} \times 2R}$ and $\mathbf{b}_p \in \mathbb{R}^{2R}$ are the weights for the entity proposal layer; $\mathbf{s}_k$ contains $2R$ scores including $R$ scores for being an entity and R scores for not being an entity at position $k$.

In the Proposal Network, we employ the cross-entropy loss as follows:

$$L_p = -\sum_{k=1}^{K} \sum_{r=1}^{R} \mathbf{y}_k^r \log \mathbf{s}_k^r, \tag{5}$$

where $\mathbf{y}_k^r$ is the label for proposal type $r$ at position $k$ and $\mathbf{s}_k^r$ is the probability of being an entity for proposal type $r$ at position $k$. It is worth mentioning that, most entity proposals are negative proposals. Thus, to balance the influence of positive proposals and negative proposals in the loss function, we keep all positive proposals and use down-sampling for negative proposals when calculating the loss $L_p$. For each batch, we fix the number of the total proposals, including all positive proposals and sampled negative proposals, used in the loss function as $N_b$. In the inference procedure of the Proposal Network, an entity proposal will be recognized as an entity candidate if its score of being an entity is higher than score of not being an entity.

### 2.2 THE CLASSIFICATION NETWORK

The Classification Network aims to classify entity candidates obtained from the Proposal Network into different categories. Here we use both sentence-level context information and the proposed Entity-Context attention to help the model focus on entity-related context tokens. The framework of the Classification Network is shown in the right part of Figure 2. Basically, it consists of three modules: Word Representation, Entity Representation and Entity Classification.

### 2.2.1 WORD REPRESENTATION

The network architecture used for learning word representations in the Classification Network is the same as that in the Proposal Network. It is a concatenation of the pre-trained word-level embedding from (Pennington et al., 2014) and a character-level word information learned from a bidirectional character LSTM (Hochreiter & Schmidhuber, 1997).

### 2.2.2 ENTITY REPRESENTATION

According to the proposal type illustrated in Figure 3, we can obtain words contained in the entity candidates from the utterances. Similar to the sentence representation in the Proposal Network, we concatenate the word representation and the ELMo language model embeddings together as the entity features. A bidirectional LSTM with hidden size $D_{el}$ is applied to the entity feature to capture sequence information among the entity words. The last hidden states of the forward and backward Entity LSTMs are concatenated as the entity representation $\mathbf{e} \in \mathbb{R}^{2D_{el}}$.

### 2.2.3 ENTITY CLASSIFICATION

A same word in different context may have different semantic meanings.Thus, in our model, we take the context information into consideration when learning the semantic representations of entity candidates. We capture the context information from other words in the same utterance. Denote $\mathbf{c}$ as the context feature vector for these context words, and it can be extracted from the sentence representation in the Proposal Network. The sentence features trained in the Proposal Network is directly transferred to the Classification Network to improve the speed of model learning.

An easy way to model context words is to concatenate all the word representations or average them. However, this naive approach may fail when there exists a lot of unrelated context words. To select high-relevant context words and learn an accurate context representation, we propose an Entity-Context attention mechanism, where the goal is to simulate the relatedness between the context and the entity. The attention module takes the entity representation $\mathbf{e}$ and all the context features $\mathbf{C} = [\mathbf{c_1}, \mathbf{c_2}, ..., \mathbf{c_N}]$ as the inputs and outputs a vector of attention weights $\mathbf{a}$:

$$\mathbf{a} = softmax(\mathbf{C}\mathbf{W}\mathbf{e}^T), \tag{6}$$

where $\mathbf{W} \in \mathbb{R}^{2D_{sl} \times 2D_{el}}$ is a weight matrix for the attention layer, and $\mathbf{a}$ is the Entity-Context attention weight on different context words. To help the model focus on entity-related context, the attentive vector $\mathbf{C}^{att}$ is calculated as the attention-weighted context:

$$\mathbf{C}^{att} = \mathbf{a} * \mathbf{C}. \tag{7}$$

The lengths of the attentive context $\mathbf{C}^{att}$ varies in different contexts. However, the goal of the Classification Network is to classify entity candidates into different categories, and thus it requires a fixed embedding size. We achieve that by adding another LSTM layer. An Attention LSTM with the hidden dimension $D_{ml}$ is used and the concatenation of the last hidden states in the forward and backward LSTM layer as the context representation $\mathbf{m} \in \mathbb{R}^{2D_{ml}}$. Hence the shape of the context representation is aligned. We concatenate the context representation and the entity representation together as the feature to classify entity candidates: $\mathbf{o} = [\mathbf{m}; \mathbf{e}]$.

A two-layer fully connected neural network is used to classify candidates into pre-defined categories:

$$\mathbf{p} = \text{softmax}\left(\mathbf{W}_{c2}\left(\sigma\left(\mathbf{o}\mathbf{W}_{c1} + \mathbf{b}_{c1}\right)\right) + \mathbf{b}_{c2}\right), \tag{8}$$

where $\mathbf{W_{c1}} \in \mathbb{R}^{(2D_{ml}+2D_{el}) \times D_h}$, $\mathbf{b_{c1}} \in \mathbb{R}^{D_h}$, $\mathbf{W_{c2}} \in \mathbb{R}^{D_{c1} \times (D_t+1)}$, $\mathbf{b_{c2}} \in \mathbb{R}^{D_t+1}$ are the weights for this fully connected neural network, and $D_t$ is the number of entity types. Actually, this classification function classifies entity candidates into $(D_t + 1)$ types since we add one type for the entity candidates which are not real entities. Finally, the hinge-ranking loss is adopted in the Classification Network:

$$L_c = \sum\nolimits_{y_w \in Y_w} max\left\{0, \Delta + \mathbf{p}_{y_w} - \mathbf{p}_{y_r}\right\}, \tag{9}$$

where $\mathbf{p}_w$ is the probability for the wrong labels $y_w$, $\mathbf{p}_r$ is the probability for the right label $y_r$, and $\Delta$ is a margin. The hinge-rank loss urges the probability for the right label higher than the probability for the wrong labels and improves the classification performance.

## 3 EXPERIMENTS

### 3.1 DATASETS

To evaluate the performance of the proposed MGEPN model for NER, we conduct the experiments on two different datasets: the CoNLL-2003 dataset (Tjong Kim Sang & De Meulder, 2003) and the

OntoNotes 5.0 dataset (Hovy et al., 2006; Pradhan et al., 2013). Specifically, there are four types of named entities in the CoNLL-2003 dataset like including locations, persons, and organizations. The OntoNotes 5.0 dataset is larger and more challenging compared with the CoNLL-2003 dataset. Essentially, it contains 18 types of entities, such as persons, time, money and languages. For the Multi-grained task, we generate two datasets which contained multi-grained entities, named MG CoNLL-2003 and MG OntoNotes 5.0, based on the CoNLL-2003 dataset and the OntoNotes 5.0 dataset, respectively. We label coarse-entities that are contained in the fine-grained entities and add them to the multi-grained datasets. An overview of these four datasets are illustrated in Table 1.

Table 1: Dataset statistics. CoNLL-2003 and OntoNotes 5.0 datasets are used for the NER tasks. MG CoNLL-2003 and MG OntoNotes 5.0 datasets are used for the Multi-grained NER tasks.

| Dataset | CoNLL-2003 | MG CoNLL-2003 | OntoNotes 5.0 | MG OntoNotes 5.0 |
|---|---|---|---|---|
| # Training Entities | 23,499 | 25,006 | 81,828 | 128,738 |
| # Validation Entities | 5,942 | 6,272 | 11,066 | 20833 |
| # Test Entities | 5,648 | 5,986 | 11,257 | 12971 |
| Word Vocab Size | 22,216 | | 50,394 | |
| Character Vocab Size | 83 | | 117 | |
| Entity Types | 4 | | 18 | |

## 3.2 IMPLEMENTATION DETAILS

We employ the Adam optimizer (Kingma & Ba, 2014) with learning rate decay for all the experiments. The learning rate is set as 0.001 at the beginning and exponential decayed by multiply 0.9 after each epoch. The batch size is set to 20 for CoNLL-2003 and 10 for Ontonotes 5.0, respectively. To alleviate over-fitting, we add dropout regularizations after all LSTM layers in our proposed MGEPN model with a dropout rate 0.5. In addition, we employ the early stopping strategy when there is no performance improvement on the development dataset after three epochs. The pre-trained word embeddings are from GloVe (Pennington et al., 2014), and the word embedding dimension $D_w$ is 300. Besides, the ELMo 5.5B data [1] is utilized in the experiment for the language model embedding. Moreover, the character embedding size is 100, and the hidden size of the Character LSTM $D_{cl}$ is also 100. The hidden size of the Entity LSTM $D_{el}$ and the Attention LSTM $D_{ml}$ are 300, respectively. We set the maximum length of entity proposals $R$ to 5. The hidden dimension of the classification layer $D_h$ is 50. The margin $\Delta$ in the hinge-ranking loss for the entity category classification is set to 5. The hidden dimension of the Word LSTM layer $D_{wl}$ is 300 for CoNLL-2003 and 1024 for OntoNotes 5.0. The hidden dimension of the sentence LSTM layer $D_{sl}$ is 300 for CoNLL-2003 and 128 for OntoNotes 5.0. The ELMo scale parameter $\gamma$ used in the Proposal Network is 3.35 for CoNLL-2003 and 3.05 for OntoNotes 5.0. The ELMo scale parameter $\gamma$ used in the Classification Network is 3.05 for CoNLL-2003 and 2.95 for OntoNotes 5.0, respectively.

## 3.3 RESULTS

**Multi-grained NER Task**. The advantage of the proposed MGEPN is to detect multi-grained named entities. In order to validate this advantage, we compare MGEPN with a classical sequence labeling model LSTM-LSTM-CRF for the Multi-grained NER task. The LSTM-LSTM-CRF baseline combines LSTM with CRF. It uses one bidirectional LSTM to capture character information and another bidirectional LSTM to capture sentence information.

Experiment results of the Multi-grained NER task on the MG CoNLL-2003 dataset and the MG OntoNotes 5.0 dataset are reported in Table **??**. We can observe from Table **??** that, our proposed model MGEPN achieves significant improvements compared with the baseline approach. For both two datasets, our model outperforms the basic LSTM-LSTM-CRF model for more than 10% in terms of precision, recall, as well as the F1 score. For sentences with over-lapping entities, we generate a sequence label for each annotation and feed them to sequence labeling based methods during both training and testing.

The first step in MGEPN is to propose entity candidates in the Proposal Network, where the effectiveness of proposing correct entity candidates immediately affects the performance of the whole model. To this end, we provide the experiment results of proposing multi-grained named entities

---

[1] https://allennlp.org/elmo

Table 2: Multi-grained NER performance on test set for MG CoNLL2003, MG OntoNotes 5.0 and ACE2005.

| | MG CoNLL2003 | | | MG OntoNotes 5.0 | | | ACE2005 | | |
|---|---|---|---|---|---|---|---|---|---|
| | P | R | F1 | P | R | F1 | P | R | F1 |
| Lample et al. (2016) | 79.38 | 77.68 | 78.52 | 60.07 | 76.66 | 67.36 | 64.05 | 52.38 | 57.63 |
| Xu et al. (2017) | 87.99 | 77.55 | 82.44 | 71.61 | 69.59 | 70.58 | 67.4 | 55.1 | 60.6 |
| Katiyar & Cardie (2018) | - | - | - | - | - | - | 70.6 | 70.4 | 70.5 |
| Ju et al. (2018) | 88.19 | 82.67 | 85.34 | 84.40 | 71.17 | 77.22 | 74.2 | 70.3 | 72.2 |
| Wang & Lu (2018) | **95.05** | 81.78 | 87.92 | **85.35** | 74.64 | 79.64 | 76.8 | **72.3** | 74.5 |
| MGEPN | 90.06 | **91.55** | **90.80** | 77.13 | **86.30** | **81.46** | **81.93** | 68.75 | **74.76** |

in Table 3. As shown in Table 3, on the MG CoNLL-2003 dataset, MGEPN obtains pretty high F1 score for proposing multi-grained named entities on both the development set and the test set.

Table 3: Performance for the Proposal Network in the Multi-grained NER task.

| **Dataset** | **Dev** | | | **Test** | | |
|---|---|---|---|---|---|---|
| | Pre. | Rec. | F1 | Pre. | Rec. | F1 |
| MG CoNLL-2003 | 95.66 | 96.60 | 96.13 | 93.75 | 96.04 | 94.88 |
| MG OntoNotes 5.0 | 81.04 | 90.10 | 85.33 | 79.85 | 88.37 | 83.89 |
| ACE2005 | 87.03 | 90.30 | 88.64 | 84.95 | 89.35 | 87.09 |

**Traditional NER Task**. We also evaluate the proposed MGEPN model on traditional NER tasks comparing with a series of classical NER models to show the effectiveness of MGEPN. Specifically, those baselines include: 1) Lample et al. (2016), which adopts the LSTM-CRF structure; 2) Ma & Hovy (2016), which uses a LSTM-CNNs-CRF architecture; 3) Chiu & Nichols (2016), which proposes a CNN-LSTM-CRF model; 4) Peters et al. (2017), which adds semi-supervised language model embeddings; and 5) Peters et al. (2018), which utilizes the state-of-the-art ELMo language model embeddings.

Table 4: F1 scores for the English NER task on the CoNLL-2003 dataset and the OntoNotes 5.0 dataset. Mean and standard deviation across five runs are reported.

| **Model** | **CoNLL-2003** | | **OntoNotes 5.0** | |
|---|---|---|---|---|
| | **Dev** | **Test** | **Dev** | **Test** |
| Xu et al. (2017) | - | 90.85 | - | - |
| Lample et al. (2016) | - | 90.94 | - | - |
| Ma & Hovy (2016) | 94.74 | 91.21 | - | - |
| Chiu & Nichols (2016) | $94.03 \pm 0.23$ | $91.62 \pm 0.33$ | $84.57 \pm 0.27$ | **$86.17 \pm 0.22$** |
| Peters et al. (2017) | - | $91.93 \pm 0.19$ | - | - |
| Peters et al. (2018) | - | $92.22 \pm 0.10$ | - | - |
| MGEPN | **$95.24 \pm 0.13$** | **$92.28 \pm 0.12$** | **$85.24 \pm 0.20$** | $82.37 \pm 0.17$ |
| MGEPN w/o context | $95.21 \pm 0.12$ | $92.23 \pm 0.06$ | $85.06 \pm 0.19$ | $82.11 \pm 0.24$ |
| MGEPN w/o attention | $95.23 \pm 0.06$ | $92.26 \pm 0.09$ | $85.13 \pm 0.24$ | $82.20 \pm 0.25$ |

Table 4 shows the F1 scores of different approaches for NER on two experimental datasets. It can be observed from Table 4 that the proposed MGEPN model outperforms all the baselines on the CoNLL-2003 dataset and achieves comparable performance on the OntoNotoes 5.0 dataset. To study the contribution of different modules in MGEPN, we also report the performance of two reduced variations of the proposed MGEPN at the bottom of Table 4. MGEPN w/o attention is a variation of MGEPN which reducing the Entity-Context attention mechanism, and MGEPN w/o context is another variation which reduces all the context information. By purely adding the context information, the F1 score on the CoNLL-2003 test set improves from 92.23 to 92.26, and by adding the attention mechanism, the F1 score improves to 92.28. Also, we show the performance for proposing entity candidates in the NER task in Table 5.

Table 5: Performance for the Proposal Network in the traditional NER task.

| Dataset | Dev | | | Test | | |
|---|---|---|---|---|---|---|
| | Pre. | Rec. | F1 | Pre. | Rec. | F1 |
| CoNLL-2003 | 96.74 | 96.41 | 96.57 | 95.33 | 95.69 | 95.51 |
| OntoNotes 5.0 | 85.69 | 90.60 | 88.08 | 82.26 | 89.43 | 85.70 |

## 3.4 CASE STUDY

To demonstrate the benefit of the proposed MGEPN, we conduct the following case study. We first select three utterances that contain multi-grained entities from the MG OntoNotes 5.0 dataset, then run the proposed MGEPN, and finally show the results in Figure 4. In Figure 4, fine-grained entities are marked with red boxes like "Academia Sinica 's institute of Biomedical Science", and coarse-grained entities are highlighted in yellow like "the Financial Times". The baseline model LSTM-LSTM-CRF only recognizes one coarse-grained entity at the selected area, but the proposed MGEPN model can successfully detect both entities in each example.

Dr. Konan Peck , an assistant research fellow at Academia Sinica 's Institute of Biomedical Sciences ...

.... insurance services for the overseas business of the Gansu Province Thermoelectric Company

James Drummond , correspondent for the Financial Times of London speaking to us from Cairo .

Figure 4: Examples from the Multi-grained OntoNotes 5.0 dataset. Fine fine-grained entities are marked with red boxes and coarse-grained entities are highlighted in yellow.

## 4 RELATED WORK

Existing approaches for recognizing named entities usually cast the NER task to the problem of sequence labeling, and thus several sequence labeling models are proposed, including linear statistical models such as Conditional Random Fields (CRF) (Ratinov & Roth, 2009), and deep neural networks like recurrent neural networks or convolutional neural networks (CNN). (Hammerton, 2003) is the first work to use LSTM for NER. Collobert et al. (2011) employ a CNN-CRF structure, which obtains competitive results to statistical models. Santos & Guimaraes (2015) introduce a character CNN to augment the CNN-CRF model. Most recent work leverages an LSTM-CRF architecture. Huang et al. (2015) use hand-crafted spelling features; Ma & Hovy (2016) and Chiu & Nichols (2016) utilize a character CNN to represent spelling characteristics; Lample et al. (2016) employ a character LSTM instead. The attention mechanism is also introduced to NER to dynamically decide how much information to use from a word or character level component (Rei et al., 2016).

External resources have been used to further improve the NER performance including Peters et al. (2017) and semi-supervised learning. (Peters et al., 2017) adds pre-trained context embeddings from bidirectional language models to NER. ELMo (Peters et al., 2018) learns a linear combination of internal hidden states stacked in a deep bidirectional language model to utilize both higher-level states which capture context-dependent aspects and lower-level state which model aspects of syntax.

Other directions for NER related work are transfer learning and multi-task learning. Transfer learning models for NER transfer information from a source task with plentiful annotations to improve the performance on a target task (Luan et al., 2018; Yang et al., 2017; Lee et al., 2017a; Zirikly & Hagiwara, 2015). These models can be transferred between different datasets or even different languages. Multi-task learning models jointly learn the NER task with other tasks like intent prediction (Goo et al., 2018) to obtain better semantic frame results by the global optimization, or fine-grained named entity categorization (Aguilar et al., 2017) for more generalized feature representations.

Several works (Xu et al., 2017; Katiyar & Cardie, 2018; Ju et al., 2018; Wang & Lu, 2018) have been proposed to address the task of nested NER where entities may have overlaps. To tackle this issue, we propose MGEPN, which can accurately detect both fine-grained and coarse-grained entities by first proposing entity candidates with the designed Proposal Network and then classifying entities using the proposed Classification Network.

## 5 CONCLUSIONS

In this work, we propose a novel Named Entity Recognition approach, called Multi-grained Entity Proposal Network (MGEPN), which consists of two modules: the Proposal Network and the Classification Network. The Proposal Network aims to output all correct entity candidates. Among these candidates, both fine-grained and coarse-grained entities are proposed, which enables the proposed MGEPN to identify multi-grained entities. For better understanding the entities, context information learned in the Proposal Network is then transferred to the Classification Network. In the Classification Network, We develop a novel Entity-Context attention mechanism to help the model focus on entity-related context information. Experiments on two real-world datasets show the effectiveness of the proposed approach for both the NER task and the Multi-grained NER task.

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
