# OpenReview forum: "Multi-Grained Entity Proposal Network for Named Entity Recognition"
_ICLR.cc/2019/Conference_

### Official Review · AnonReviewer3 · 2018-11-02
**Nested NER dection**

**Rating:** 4
**Confidence:** 4

**Review:**

This paper describes multi-grained entity recognition. Experimental results show that the proposed Multi-Grained Proposal Network achieve better performance on NER tasks.

Major comments:

- A major weakness of this paper is lack of citations to recent related studies. There are studies on nested NER published this June:

A. Katiyar and C. Cardie, Nested Named Entity Recognition Revisited, NAACL/HLT 2018, June, 2018.
M. Ju, et al., A neural layered model for nested named entity recognition, NAACL/HLT 2018, June 2018.

You need to compare these conventional methods to your proposed method.

---

> ### Author Response · Authors · 2018-11-26
> **Comments on Review**
>
> Thanks a lot for your review.
>
> We’ve updated our table with baselines [1,2,3] in the revised paper.
>
> This is a brief version of Multi-grained NER F1 Performance on the test sets of three datasets:
>
> 						  MG CoNLL2003      	MG OntoNotes 5.0 		 ACE 2005
> Lample et al. (2016)	     |          78.52		 |		  67.36               |    	57.63            |
> Xu et al. (2017)		     |          82.44		 |		  70.58               |    	60.6              |
> Katiyar & Cardie (2018)    | 	    -		 	 | 	             -	                  |		70.5              |
> Ju et al. (2018)                    |         85.34                |               77.22               |           72.2              |
> Wang & Lu (2018)             |          87.92               |               79.64               |           74.5              |
> -------------------------------------------------------------------------------------------------------------------------------
> MGEPN				    |          90.80                |               81.46                |         74.76            |
>
> [1] A. Katiyar and C. Cardie, Nested Named Entity Recognition Revisited, NAACL/HLT 2018, June, 2018.
> [2] M. Ju, et al., A neural layered model for nested named entity recognition, NAACL/HLT 2018, June 2018.
> [3] Wang et al., Neural Segmental Hypergraphs for Overlapping Mention Recognition, EMNLP 2018, Nov 2018.

---

### Official Review · AnonReviewer2 · 2018-11-03
**Interesting task of multi-grained NER, reasonable models.**

**Rating:** 5
**Confidence:** 3

**Review:**


<Summary>
Authors propose the “Multi-grained NER (MGNER)  task” which aims at detecting entities at both coarse and fine-grained levels. Authors propose a Multi-grained Entity Proposal Network (MGEPN) which comprises (1) a Proposal Network that determines entity boundaries, and (2) a Classification network that classifies each proposed segment of an entity.

The task is primarily tested against the proposed method itself. The proposed method does outperform traditional sequence-labeling baseline model (LSTM-LSTM-CRF), validating the proposed approach. When the proposed model (trained with extra MG data) is evaluated on the traditional NER task (on test sets), however, no significant improvement is observed -- I believe this result is understandable though, because e.g. MG datasets have slightly different label distributions from original datasets, hence likely to result in lower recall, etc.

<Comments>
The task studied is interesting, and can potentially benefit other downstream applications that consume NER results -- although it seems as though similar tasks have been studied prior to this study. The novelty of the proposed architecture is moderate - while each component of the model does not have too much technical novelty, the idea of separating the model into a proposal network and a classifier seems to be a new approach in the context of NER (that diverges from the traditional sequence labelling approaches), and is reasonably designed for the proposed task.

The details for creating the MG datasets is missing - are they labeled by human labelers, or bootstrapped? Experts or crowd-sourced? By how many people? Will the new datasets be released? Please provide clarifications.

The proposed approach does not or barely outperform base models when tested on the traditional NER task -- the proposed work thus can be strengthened by better illustrating the motivation of the MGNER task and/or validating its efficacy in other downstream tasks, etc.

Authors could provide better insights into the new proposed task by providing more in-depth error analysis - especially the cases when MG NER fails as well (e.g. when coarse-grained prediction predicts a false positive named-entity, etc.)

---

> ### Author Response · Authors · 2018-11-26
> **Comments on Review**
>
> Thanks a lot for your review.
>
> 1, The details for creating the MG datasets:
>
> First, we find out all the entities labeled in the original dataset such as the CoNLL2003 dataset. Then we output all the long entities which have overlappings with other known entities. Invalid entity pairs are filtered by a manual check by humans. In this way, we obtain a dictionary which stores long entities and it's containing short entities. Using this entity-containing dictionary, we labeled these short entities in the sentences where the long entities are labeled. The whole dataset cannot be released directly due to the copyright issues on the original CONLL2003 dataset and OntoNotes 5.0 dataset. However, we can release the entity-containing dictionary and the pipeline scripts. Through data processing, the MG datasets can be easily recovered if the original CONLL2003 dataset and OntoNotes 5.0 dataset are accessible.
>
> 2, Traditional NER task is addressed by sequence labeling models. However, entities in the same sentence may have overlaps. Our proposed model is able to overcome the weakness of sequential labeling schema and recognize overlapping entity recognition by proposing entities first and then classifying them into different categories.
>
> 3, Bad cases where MG NER fails:
>
> “Of course , it was discovered that his men 's basketball coach had directly given six grand to a Yugoslavian recruit , but that was just his stipend for travel and coach Jim O'Brien was just having a problem with the currency exchange rates ... right Andy ?”, is a test instance in the ACE2005 dataset. Our proposed model recognized "right Andy" as an entity with type PERSON, however the ground truth for "right Andy" is not an entity in this sentence. “Andy” is PERSON but “right Andy” is not. "right" might be recognized as an adjective which describes the qualities or states of "Andy". So additional information like the characteristic or property of words may help us recognize the correct entities.

---

### Official Review · AnonReviewer1 · 2018-11-05
**Lack of comprehensive related work and lack of clarity in the writing**

**Rating:** 5
**Confidence:** 4

**Review:**

This paper proposed a entity proposal network for named entity recognition which can be effectively detect overlapped spans. The model obtain good performance on both Multi-grained NER task and traditional NER task. The paper is in general well written, the idea of proposal network break the traditional framework of sequence tagging formulation in NER task and thus can be effectively applied to detect overlapped named entities.

However, I still have many concerns regarding the notation, the novelty of the paper, and the comparison with related literature, especially on previous overlapped span detection NER papers. The detailed concerns and questions are as follows:
The notations are very confusing. Many of the notations are not defined. For example, what does $T$ in $2D_sl*2T$ below Eq. 4 indicates?  What does $R$ scores means? I guess $R$ does not equal to number of entity types, but I’m not sure what $R$ exactly indicates. If $R$ is not number of entity types, why do you need R scores for being an entity and R scores for not being an entity? And what is $t$ in Eq 5? Is that entity type id or something else?
I’m still confused how you select the entity spans from a large number of entity candidates. In Figure 5, if the max window length is 5, there may be more span candidates than the listed 5 examples, such as t_3 t_4 t_5. How do you prune it out?
Table 5 is weird. There is not comparison with any baselines but just a report of the performance with this system. I don’t know what point this table is showing.
This is not the first paper that enumerates all possible spans for NER task.The idea of enumerating possible spans for NER task has appeared in [1] and can also effectively detect overlapped span. I would like to see the performance comparison between the two systems. The enumerating span ideas has been applied in many other tasks as well such as coreference resolution [2]and SRL[3], none of which is mentioned in related work.
I feel that most of the gain is from ELMo but not the model architecture itself, since in Table 4, the improvement from the ELMo is only 0.06. The LSTM-LSTM-CRF is without adding ELMo, which is not a fair comparison.
The comparison of baselines is not adequate and is far from enough. The paper only compares with LSTM+CRF frameworks, which are not designed for detecting overlapped spans. There are many papers on detecting overlapping spans, such as [4], [5] and [6]. It’s important to compare with those paper since those methods are especially designed for overlapped span NER tasks.
[1] Multi-Task Identification of Entities, Relations, and Coreferencefor Scientific Knowledge Graph Construction, EMNLP 2018
[2] End-to-end neural coreference resolution, EMNLP 2017
[3] Jointly predicting predicates and arguments in neural semantic role labeling, ACL 2018
[4] Nested Named Entity Recognition Revisited, NAACL 2018
[5] A Neural Layered Model for Nested Named Entity Recognition, NAACL 2018
[6] Neural Segmental Hypergraphs for Overlapping Mention Recognition, EMNLP 2018

---

> ### Author Response · Authors · 2018-11-26
> **Comments on Review**
>
> Thanks a lot for your review.
>
> 1, For the notations:
>
> 1) $T$ in $2D_sl*2T$ below Eq. 4 should be updated as $R$.
> 2) $R$ is the max length of an entity proposal. In order to limit the number of generated proposals, we set the maximum length of an entity proposal to R.
> 3) $t$ in Eq 5 should be updated as $r$.
>
> We've updated these typos in the revised version.
>
> 2, For generating proposals:
> 	The max length of an entity proposal is set as R. We use a sliding window to generate entity proposals that revolve around each token position. For each token position k, we will generate R entity proposals with length varies from 1 to R.
>
> 	For example, we have an utterance with "t1, t2, t3, t4, t5" and R is set as 5. We will generate 5 proposals revolving around t1, 5 proposals revolving around t2, 5 proposals revolving around t3, ..., 5 proposals revolving around t5. The way five proposals are devised guarantee that each candidate entity within the max length will be captured by one of the proposals at a certain token position. Figure 3 shows an example of the five proposals generated at t3.
> 	We also generate proposals for other token positions using the same strategy.
> 	For t1, we will generate 5 proposals: (t1),(t1,t2),(t0,t1,t2),(t0,t1,t2,t3),(t-1,t0,t1,t2,t3).
> 	For t2, we will generate 5 proposals: (t2),(t2,t3),(t1,t2,t3),(t1,t2,t3,t4),(t0,t1,t2,t3,t4).
> 	For t4, we will generate 5 proposals: (t4),(t4,t5),(t3,t4,t5),(t3,t4,t5,t6),(t2,t3,t4,t5,t6).
> 	For t5, we will generate 5 proposals: (t5),(t5,t6),(t4,t5,t6),(t4,t5,t6,t7),(t3,t4,t5,t6,t7).
>
> 	Proposals that contain invalid indexes like (t0,t1,t2), (t5, t6) will be deleted. Hence we can obtain all the valid entity proposals under the condition that the max length is R.
>
> 3, The R in this sentence, "s_k contains 2R scores including R scores for being an entity and R scores for not being an entity at position k", is a number. We use a two-class softmax function to determine the quality of an entity proposal. For each proposal, we will have one score for being an entity and one score for not being an entity. And there are R entity proposals at position k. So, we have 2R scores for each token position at k and s_k is a vector with dimension 2*R.
>
> 4, Our system is composed of two modules: the Proposal Network and the Classification Network. The Proposal Network generates proposals on possible entity candidates, where the classification network determines the entity type for each entity candidate. To this end, we provide the experiment results to evaluate the performance of the proposal network on the multi-grained ner task in Table 3 and the performance on the traditional ner task in Table 5.
>
> 5, We’ve updated our table with baselines [4, 5, 6] in the revised paper. [1] is proposed as a multi-task learning problem, hence we didn't compare with this model. Works like [1,2,3] are updated in the related work.
>
>
> This is a brief version of Multi-grained NER F1 Performance on the test sets of three datasets:
>
> 						  MG CoNLL2003      	MG OntoNotes 5.0 		 ACE 2005
> Lample et al. (2016)	     |          78.52		 |		  67.36               |    	57.63            |
> Xu et al. (2017)		     |          82.44		 |		  70.58               |    	60.6              |
> Katiyar & Cardie (2018)    | 	    -		 	 | 			                  |		70.5              |
> Ju et al. (2018)                    |         85.34                |               77.22               |           72.2              |
> Wang & Lu (2018)             |          87.92               |               79.64               |           74.5              |
> -------------------------------------------------------------------------------------------------------------------------------
> MGEPN				    |          90.80                |               81.46                |         74.76            |
>
> [1] Multi-Task Identification of Entities, Relations, and Coreference for Scientific Knowledge Graph Construction, EMNLP 2018
> [2] End-to-end neural coreference resolution, EMNLP 2017
> [3] Jointly predicting predicates and arguments in neural semantic role labeling, ACL 2018
> [4] Nested Named Entity Recognition Revisited, NAACL 2018
> [5] A Neural Layered Model for Nested Named Entity Recognition, NAACL 2018
> [6] Neural Segmental Hypergraphs for Overlapping Mention Recognition, EMNLP 2018

---

### Meta-Review · Area_Chair1 · 2018-12-14

**Confidence:** 4
**Recommendation:** Reject

**Metareview:**

The authors present a method for fine grained entity tagging, which could be useful in certain practical scenarios.

I found the labeling of the CoNLL data with the fine grained entities a bit confusing.  The authors did not talk about the details of how the coarse grained labels were changed to fine grained ones.  This detail is important and is missing from the paper.  Moreover, there are concerns about the novelty of the work, both in terms of the task definition and the model (see the review of Reviewer 1, e.g.).

There is consensus amongst the reviewers, in that, their feedback is lukewarm about the paper.